# Heart Involvement in Inflammatory Rheumatic Diseases: A Systematic Literature Review

**DOI:** 10.3390/medicina55060249

**Published:** 2019-06-06

**Authors:** Florina Buleu, Elena Sirbu, Alexandru Caraba, Simona Dragan

**Affiliations:** 1Departament of Cardiology, Faculty of Medicine, University of Medicine and Pharmacy “Victor Babeș”, Timișoara 300041, Romania; buleu.florina@gmail.com (F.B.); simona.dragan@umft.ro (S.D.); 2Department of Physical Therapy and Special Motricity, West University of Timișoara, Timișoara 300223, Romania; 3Departament of Internal Medicine, Faculty of Medicine, University of Medicine and Pharmacy “Victor Babeș”, Timișoara 300041, Romania; alexcaraba@yahoo.com

**Keywords:** heart, systemic inflammation, autoimmunity, rheumatic diseases, disease activity

## Abstract

*Introduction*: Patients with inflammatory rheumatic diseases have an increased risk of developing cardiovascular manifestations. The high risk of cardiovascular pathology in these patients is not only due to traditional cardiovascular risk factors (age, gender, family history, smoking, sedentary lifestyle, cholesterol), but also to chronic inflammation and autoimmunity. *Aim:* In this review, we present the mechanisms of cardiovascular comorbidities associated with inflammatory rheumatic diseases, as they have recently been reported by different authors, grouped in electrical abnormalities, valvular, myocardial and pericardial modifications and vascular involvement. *Methods*: We conducted a systematic search of published literature on the following online databases: EBSCO, ScienceDirect, Scopus and PubMed. Searches were limited to full-text English-language journal articles published between 2010 and 2017 using the following key words: heart, systemic inflammation, autoimmunity, rheumatic diseases and disease activity. After the primary analysis we included 50 scientific articles in this review. *Results*: The results showed that cardiac manifestations of systemic inflammation can occur frequently with different prevalence in rheumatoid arthritis (RA), systemic lupus erythematosus(SLE), systemic sclerosis(SSc) and ankylosing spondylitis(AS). Rheumatologic diseases can affect the myocardium, cardiac valves, pericardium, conduction system and arterial vasculature. *Conclusions*: Early detection, adequate management and therapy of specific cardiac involvement are essential in rheumatic disease. Electrocardiographic and echocardiographic evaluation should be performed as routine investigations in patients with inflammatory rheumatic diseases.

## 1. Introduction

Inflammatory rheumatic diseases are associated with an increased risk of developing several cardiovascular comorbidities [1]. The increased cardiovascular disease risk in patients with systemic rheumatic diseases is conditioned, partially, by the presence of cardiovascular risk factors (age, gender, family history, smoking, sedentary lifestyle, dyslipidemia). Genetic risk has also been shown to play a substantial role in global cardiovascular risk [2]. Risk factors associated with or modified by rheumatic diseases have an important role in cardiovascular involvement [3]. Stratification of cardiovascular risk in these patients by using risk diagrams is therefore essential [4]. Inflammation is crucial for vascular dysfunction, affecting patients with inflammatory autoimmune diseases all the more. Chronic inflammation and autoimmunity may lead to accelerated atherosclerosis. Markers of high disease activity or disease severity scores are linked to increased cardiovascular risk [1].

Autoimmune-inflammatory rheumatic diseases, such as rheumatoid arthritis (RA), systemic lupus erythematosus (SLE), systemic sclerosis (SSc) and ankylosing spondylitis (AS) have been associated with accelerated atherosclerosis leading to increased cardiovascular risk. Inflammatory cells, chemokines, cytokines, proteases, autoantibodies and adhesion receptors are involved in this process, which can directly affect all structures of the cardiovascular system, such as the myocardium, cardiac valves, pericardium, conduction system and vasculature [5]. Cardiovascular involvement may be mild and clinically silent, but in some cases can become severe and life threatening. Increased morbidity and mortality warrants early diagnosis and treatment [3].

### Objective

We used a systematic review methodology to assess recent literature on the involvement of the heart in inflammatory rheumatic diseases, grouped in electrical abnormalities, valvular, myocardial, pericardial and vascular involvement.

## 2. Materials and Methods

Searches were conducted in the following online databases: EBSCO, ScienceDirect, Scopus and PubMed. They were limited to full-text English-language journal articles published between 2010 and 2017 using the following key words: heart, systemic inflammation, autoimmunity, rheumatic diseases and disease activity.

Assessment of studies was done independently by two blinded reviewers. Disagreements between them were resolved by consensus using predefined eligibility criteria. Two levels of screening were applied. At level 1, titles and abstracts were reviewed to exclude irrelevant studies; at level 2, full-text articles were reviewed to determine the relevance of the studies.

A study was included if (a) the abstract was available, (b) it contained original data, (c) it used accepted classification criteria for each rheumatic disease, (d) it discussed CV risk factors (traditional and/or nontraditional) and (e) it examined cardiovascular involvement (electrical abnormalities, valvular, myocardial, pericardial and vascular). Articles were excluded if they were case reports, if they discussed topics not related to cardiovascular involvement in rheumatic diseases, if they did not meet the inclusion criteria, if they had insufficient data or if they had results that showed lack of statistical significance.

## 3. Results

The search of electronic databases resulted in 95 articles selected based on titles and abstracts. From these, 68 full-text articles were assessed for eligibility criteria (Figure 1). After reviewing the full text and removing duplicates 64 full-text articles remained, but only 54 studies met the predefined inclusion criteria. Four studies were excluded because they did not meet the inclusion criteria. Hence, 50 relevant studies were selected in this systematic review.

## 4. Discussion

We examined cardiovascular involvement in the following rheumatic diseases: rheumatoid arthritis, systemic lupus erythematosus, systemic sclerosis and ankylosing spondylitis.

### 4.1. Electrical Abnormalities 

Electrical abnormalities were represented by sudden cardiac death (SCD), ventricular arrhythmia, supraventricular tachycardia and atrioventricular block, predominantly affecting patients with inflammatory rheumatic diseases [6]. In patients with rheumatoid arthritis, acute coronary syndromes and consequent ventricular arrhythmias have been reported as main cause of SCD due to accelerated atherosclerotic coronary artery disease [6]. The risk of electric rhythm and conduction disturbances and consequent SCD was higher in RA patients compared to controls [6]. Although some studies have reported that many invasive and non-invasive techniques were not useful to identify patients at risk of ventricular arrhythmias [7], analysis of heart rate variability proved to have a prognostic significance in these patients [6]. A significant negative correlation was observed between disease activity and heart rate variability in RA patients [8]. In patients with high activity of RA, the decrease of heart rate variability reflects severity of inflammation and the latter predicts an increased risk for ventricular arrhythmias, SCD or acute myocardial infarction [6,9].

In contrast to rhythm abnormalities, conduction disorders proved to be more common in inflammatory rheumatic disorders. In an older study, Villecco et al. noted that the detection of antibodies against conductive tissue, mainly the atrioventricular (AV) node, did cause right bundle branch block in 35% of 60 patients with RA [10]. According to Wisłowska et al., cardiac impairment occurred more commonly in nodular than in non-nodular RA or control patients. Furthermore, they found more patients with less 1 mm ST-segment depression on 24h EKG Holter monitoring, compared with individuals in the control group [11].

Cardiac involvement in rheumatoid arthritis was also analyzed in a French study done on 106 patients with RA (mean duration of disease, 12 ± 9 years), and 74 control patients with degenerative joint disease. Repolarization defects and negative T waves occurred more frequently (21%) in RA patients compared to control [12,13].

Rhythm disorders can occur among patients with SLE. Arrhythmias, which are reported more often, include sinus tachycardia, atrial fibrillation and ectopic atrial beats (associated with SLE flares and SLE myocarditis) [14]. Conversely, malignant ventricular arrhythmias are rarely reported in SLE patients. So far, a consensus on the prevalence of arrhythmias in SLE patients does not exist due to the small number of studies in the literature. However, sinus tachycardia was reported in 50% of patients, and may be the only cardiac manifestation of SLE resolved under corticosteroid treatment [6]. Other rhythm abnormalities like sinus bradycardia and QT interval prolongation are associated with high titers of anti-small cytoplasmic ribonucleoprotein antibodies (anti-Ro/SSA) [6].

Conduction disturbances in SLE were shown to be due to small vessel vasculitis and infiltration of fibrous or granulation tissue. Conduction defects may also appear as a sequel of myocarditis in 34–70% of patients with SLE, but first-degree heart block is often transient [6].

Another study conducted by Wislowska et al. assessed the systolic and diastolic function of the left ventricle (LV) by echocardiography in SLE patients with no clinically evident cardiovascular disease compared to controls, correlating the findings with duration and severity of SLE. Results showed no statistically significant correlation for systolic or diastolic function, except for fractional shortening, especially in patients with more than 10 years of disease duration (*p* < 0.005) and a Systemic Lupus Erythematosus Disease Activity Index (SLEDAI) higher than six points (*p* < 0.01). In SLE subjects with long disease duration, left atrial end-systolic diameter was significantly greater (*p* < 0.05) and the ejection fraction was lower (*p* < 0.05) than in controls [15].

Lazzerini et al. observed that conduction defects happen more frequently in anti-Ro-positive than in anti-Ro-negative SLE patients [16].

Atrial fibrillation, flutter or paroxysmal supraventricular tachycardia were reported in 20–30% of systemic sclerosis patients. A large number of these patients (up to 67%) had ventricular arrhythmias assessed by electrocardiography. The most common arrhythmia was represented by premature ventricular beats associated with a risk of 50% mortality and SCD. Ambulatory 24 h Holter electrocardiography recorded bundle and fascicular blocks in 25–75% of cases, and very rarely second- and third-degree AV blocks (<2%) [6].

The increased risk of conduction disturbances is related to fibrosis of the sinus node and bundle branches, but direct involvement of cardiac conduction tissue and its arterial blood supply has also been reported [6].

Inankylosing spondylitis conduction, disturbances may develop as a consequence ofthe inflammatory status of the myocardial tissue in 2–20% of cases. Although first-degree AV block is most common, higher grade blocks and right and left bundlebranch blocks have also been reported. In addition, heart blocks appear more frequently in HLA-B27–positive patients, even in the absence of joint disease. Atrial fibrillation has also been reported in AS, especially in HLA-B27 positive patients [17].

### 4.2. Valvular Involvement

Valvular involvement includes valvular regurgitation, valvular nodules and Liebman–Sacks vegetations. Valvular diseases are common among patients with inflammatory rheumatic diseases, especially in patients with RA, SLE, antiphospholipid antibodies syndrome or AS.

Valvular disease was the most common cardiac abnormality confirmed by echocardiography or in autopsies, present in 30% of patients with rheumatoid arthritis, but in most of the cases it was asymptomatic. Several studies showed that mitral regurgitation was the most common form of valvular disease in RA patients [17,18].

Using transesophageal echocardiography (TEE), Guedes et al. reported the presence of mitral regurgitation in 80% of patients with RA compared to 37% in the control population. In contrast, the prevalence of aortic and tricuspid regurgitation was not different in RA patients compared to the control. The most commons detected lesions were valve thickening, valve prolapse and nodules [19]. Moreover, cardiac involvement is a common and significant cause of morbidity and mortality in systemic lupus erythematosus patients. Both endocarditis and the presence of valve nodules were described in SLE patients, and TEE revealed that more than 50% of patients have valvular abnormalities [3]. Endocarditis, frequently present in SLE, also known as Liebman–Sacks endocarditis, is more commonly detected in echocardiographic studies than in clinical settings. In a study performed on 342 consecutive patients with SLE, Moyssakis et al. reported Libman–Sacks vegetations in 1 of 10 patients. Their incidence increased with SLE duration and disease activity [20]. The severity of valvular regurgitation in SLE patients may be related to high levels of IgG anticardiolipin antibodies [18,21].

In contrast, valvular disease is rare in systemic sclerosis (SSc) patients. Thickening of the aortic and mitral valves with regurgitation was first noted by Kinney et al. [22]. Such lesions were found in 18% of autopsied SSc patients (5 out of 28 cases), and lesions of the tricuspid valves were also detected, in combinations or alone [23].

Echocardiography performed in a study by Qazi Masood et al. on 47 patients with SSc revealed valvular abnormalities (alone or in combination) in 5 patients, mitral regurgitation in 2 patients, mitral stenosis in 2 patients and significant aortic regurgitation in 2 patients [24].

Aortic disease, which includes aortic regurgitation and/or aortitis, is common in ankylosing spondylitis patients. The most important valvular abnormalities described in AS are aortic root thickening and dilatation, aortic cusp thickening and retraction, aortic and mitral regurgitation [18].

However, in a study on AS patients undergoing TEE, aortic root thickening was more common than dilatation (61% vs. 25%) and dilatation was usually mild. Therefore, root thickening was associated with increased stiffness, and these abnormalities (in addition to aortic valve thickening) were associated with aortic regurgitation. Aortic root stiffness leads to hypertension or increased left ventricular afterload, and therefore to ventricular hypertrophy and diastolic dysfunction. Also, aortic root or valve lesions may lead to thromboembolic complications [25].

### 4.3. Myocardial Involvement 

Involvement of the myocardium is a common finding in rheumatic diseases, especially through inflammatory and autoimmune mechanisms [5].

The risk of myocardial dysfunction and congestive heart failure (CHF) are high in RA patients. Moreover, congestive heart failure, more than ischemic heart disease, is an important contributor to increased mortality in RA [26]. Wolfe et al. observed a high prevalence and incidence of CHF in RA patients compared to controls (no-RA subjects) [27]. The underlying mechanism of heart failure in patients with RA is unclear, but it may be related to both left ventricular systolic and diastolic dysfunction [5,18]. Other factors, particularly the chronic production of inflammatory cytokines, are likely to contribute to myocardial dysfunction in RA [18].

Left ventricular diastolic dysfunction in rheumatoid arthritis was found to be independent of the presence of traditional cardiovascular risk factors and is probably related to a chronic proinflammatory state. Extra-articular manifestations are more common in patients with rheumatoid arthritis and diastolic dysfunction [28]. Myocarditis is a rare cardiac manifestation in systemic lupus erythematosus, as reported in autoptic and clinical studies [29]. Most cases of lupus myocarditis are asymptomatic and can be diagnosed by transthoracic echocardiography [18,28].

Abnormalities of left ventricular structure and function have been reported among the cardiac manifestations of SLE, including increased LV wall thickness and mass, decreased LV ejection fraction and impaired diastolic volumes. There is strong evidence that the mechanisms by which SLE induces alterations in LV structure include chronic inflammatory processes leading to subclinical vasculitis, myocarditis or vascular stiffening [30].

Cardiac manifestations have been reported in approximately 15–35% of systemic sclerosis patients and in the majority they remain silent. Myocardial fibrosis is the most common cardiac manifestation of SSc. Commonly, fibrosis occurs later in the course of illness and can lead to left ventricular systolic and diastolic dysfunction. Echocardiographic studies found segmental wall motion abnormalities and impaired coronary flow reserve occurring in the absence of coronary artery disease (CAD) [31]. Pulmonary involvement is another common clinical manifestation in SSc, consisting of interstitial fibrosis and pulmonary vascular disease that lead to pulmonary arterial hypertension and associated myocardial changes [32]. Using Doppler echocardiography, Follansbee et al. found significantly lower right ventricular (RV) ejection fraction compared to control. The right ventricular function determined by radionuclide ventriculography was likewise abnormal, even in the absence of pulmonary hypertension, probably due to microvascular disease [33]. Microvascular abnormalities are frequent in patients with systemic sclerosis, and abnormalities of the large arteries have also been reported recently [18,34].

In ankylosing spondylitis, the risk of heart failure was found to be 1.34-fold greater than in controls, considering the cause of LV diastolic dysfunction [35].Similarly, a higher average of the *E*/*E*′ ratio, illustrating diastolic dysfunction, was present in patients with axial AS in a study conducted by Chen at al. Moreover, the same study demonstrated superiority of 2D speckle tracking analysis, which revealed LV systolic dysfunction despite an apparently normal diastolic function performed by conventional echocardiography [36].

### 4.4. Pericardial Involvement 

Pericarditis is the most frequent cardiac manifestation in rheumatoid arthritis patients. Between 30% and 50% of patients with severe RA have pericarditis at echocardiography or autopsy, which is usually clinically silent [5,37]. However, some patients may present symptoms of pericarditis (fever, pleuritic chest pain) or of significant pericardial effusion (dyspnea, fatigue, hypotension) [18]. Most often, pericarditis is found in rheumatoid factor positive male patients with severely nodular RA or other extra-articular features of RA [38]. Besides echocardiography, pericardial effusion may be incidentally detected on computed chest tomography or magnetic resonance imaging of the heart [18].

Pericardial disease is a diagnostic feature of systemic lupus erythematosus, and the most frequent clinical cardiovascular manifestation [32]. Several large clinical studies indicate that pericardial disease occurs in 20–50% of SLE patients [18,38]. Pericardial effusion is seen more frequently in active disease, but may be asymptomatic. Effusions are usually small, but moderate to large pericardial effusions were detected in 7% of patients in one series of SLE patients [32]. Electrocardiographic modifications, notably of the T wave, are the most common clinical manifestation of pericarditis in SLE [39].

Pericardial involvement has been noted in 33–72% of systemic sclerosis autopsied patients and includes fibrinous pericarditis, chronic fibrous pericarditis, pericardial adhesions and pericardial effusions [23]. Although echocardiographic studies demonstrate a higher prevalence of pericardial disease, clinically evident pericarditis was reported in only 7–20% of SSc patients. In most cases pericarditis is a primary disease, but in some cases it occurs secondary to uraemia, frequently seen in end-stage systemic sclerosis [18,23].

Pericardial effusion has rarely been reported in patients with ankylosing spondylitis, and there has only been one case of pleural and pericardial effusion association [40]. In a retrospective review of 21 cases with cardiovascular manifestations, which were based on medical records of210 cases of AS followed-up over a period of 25 years, Ben Taarit et al. reported that only one patient had pericarditis [41].

### 4.5. Vascular Involvement

One of the cardiac manifestations of rheumatoid arthritis is premature atherosclerosis, which is not necessarily correlated with traditional cardiovascular risk factors (gender, cigarette smoking, high body mass index, hypercholesterolemia, diabetes, hypertension, hypothyroidism and hyperhomocysteinemia).There are many proposed mechanisms to explain accelerated atherosclerosis in rheumatic patients: proinflammatory pathways (TNF-alpha, IL-6), adhesion molecules and cellular infiltrates, and also endothelial dysfunction, which is the first step in atherosclerosis development [3].

Many studies noted the direct relation of RA with atherosclerosis and independence from traditional risk factors [42,43,44]. Moreover, the decreased risk of cardiovascular disease in RA patients who were treated with TNF blockers and methotrexate further supports the hypothesis that accelerated atherosclerosis in RA is primarily triggered by chronic inflammation, not by traditional cardiovascular risk factors [32].

In addition, several studies have shown a two-fold increase in frequency of myocardial infarction in patients with RA when compared with controls [45]. Accelerated atherosclerosis, unstable plaques with increased chance of rupture, are mechanisms that explain the higher risk of infarction in patients with RA compared with individuals without RA [44].

Endothelial dysfunction, carotid intima-media thickness and plaque evaluations provide an accurate detection of the atherosclerotic process at a preclinical stage, before the appearance of clinical disease, allowing preventive measures to decrease cardiovascular risk in subjects with RA [46,47]. Arterial stiffness, as a preclinical marker of atherosclerosis, is a predictive factor of cardiovascular disease [48,49]. In RA, arterial stiffness is increased, being related to disease duration and to inflammatory mediators [32]. An increased prevalence of severe subclinical atherosclerosis inpatients with long-term treated rheumatoid arthritis without clinically evident atherosclerotic disease was reported for the first time by Gonzalez-Juanatey et al. [50]. Premature atherosclerosis of coronary arteries has been noted in large clinical studies on patients with systemic lupus erythematosus. Similar to other rheumatic diseases, premature atherosclerosis in lupus can result as a consequence of the disease itself, independent of traditional cardiovascular risk factors. It is considered that atherosclerotic cardiovascular disease represents an important cause of “late“ death in SLE patients [3].

In addition, patients with positive antinuclear antibodies have an increased risk of cardiovascular disease and mortality. Thus, assessments in patients with SLE focus on biological or imaging markers of subclinical atherosclerosis in order to detect early changes [1,5].

Clinical features of coronary artery disease are related to several factors including older age at onset of disease, longer duration of SLE, longer duration of treatment with corticosteroids, higher damage score, elevated levels of homocysteine and low-density lipoprotein cholesterol [32]. Moreover, Roman et al., in a study conducted on arterial stiffness in SLE patients, found that pulse wave velocity (PWV) was related to both SLE-related factors and traditional cardiovascular risk factors [51]. Results of another study have shown a positive correlation between PWV and metabolic syndrome in SLE patients [52].

Data derived from studies carried out on premature atherosclerosis in scleroderma patients produced contrasting results. A common finding was that inflammation is less prominent in SSc, resulting in less aggressive atherosclerosis and therefore making detection of subclinical atherosclerosis more difficult in these patients. The underlying mechanisms include multi-system organ inflammation, endothelial wall damage and vascular disease [53].

Circulating autoantibodies and proinflammatory cytokines present in systemic inflammation play a significant role in the development of atherosclerosis. The most common clinical manifestations in patients with SSc are represented by involvement of brain, carotid and coronary arteries with high risk of peripheral vascular disease, stroke and coronary disease [53].

SSc is characterized by microvascular abnormalities, secondary ischemia and excessive fibroblast activity. Nevertheless, recent studies have also pointed out the involvement of large arteries in patients with SSc [53]. These occur as a consequence of chronic systemic inflammation. A study conducted by Cheng et al. on 73 patients was divided into three groups: control group (*n* = 21), patients with limited cutaneous form (*n* = 33) and patients with diffuse cutaneous subset (*n* = 19). They found that carotid artery stiffness was increased in SSc patients compared with the control subjects, while stiffness of the muscular femoral artery wall was normal [34].

Considering the cardiovascular risk in patients with SSc, invasive and/or noninvasive testing is needed in order to assess vascular function, with the most important manifestation represented by microvascular abnormalities.

In a meta-analysis of observational studies, Ungprasert et al. compared coronary artery disease risk in patients with AS versus controls, and showed that the CAD risk ratio was 1:41, demonstrating an increased risk among patients with AS [54]. Aortic aneurysmal dilation is very rarely found in AS, and bicuspid aortic valve or CAD are only occasionally present. Impaired endothelial function and decreased aortic elasticity could be responsible for aortic involvement in AS patients [55].

Transthoracic echocardiography performed in 187 patients with AS showed the presence of aortic regurgitation in 18% of patients (*n* = 34), mild regurgitation in 24 patients and moderate regurgitation in 9 patients. Severe aortic regurgitation was found in 1 AS patient, and clinically significant CAD in 9 patients (5%). The same study showed that the prevalence was significantly higher than expected from population data [56].

According to the current ESC (European Society of Cardiology) guidelines, cardiovascular risk quantification in these patients is recommended every fiveyears for patients with low to moderate cardiovascular risk, because there is currently no evidence that its assessment each year reduces cardiovascular risk. In intermediary and high cardiovascular risk patients, screening should be performed more often in order to rapidly modify and lower existing risk factors. The new European League Against Rheumatism (EULAR) Guideline Recommendation includes the option of screening for carotid atherosclerotic plaques, especially in RA, due to increased association with CAD in these patients [57,58].

In studies conducted on patients with inflammatory rheumatic diseases it was observed that the risk chart algorithms used for cardiovascular risk stratification, such as SCORE (Systematic Coronary Risk Evaluation) or Framingham, underestimated the actual cardiovascular risk. In these cases, non-invasive surrogate markers such as coronary artery calcification score (CACS) and carotid ultrasonography are recommended to identify high-risk patients included in the category of moderate cardiovascular risk when applying a SCORE or the modified EULAR SCORE alone [59,60,61].

The last European Guidelines on cardiovascular disease prevention in clinical practice, recommend multiplying the risk SCORE by 1.5 to assess the CV risk in patients with autoimmune disease, because the actual risk is underestimated in these patients [62,63]. Statin trials suggest that the relative reduction in CVD incidence in patients with autoimmune diseases affirms that these drugs are indicated when the risk SCORE ≥10% (very-high-risk persons) [62].

Carotid ultrasound, echocardiography, cardiac computed tomography, positron emission tomography and cardiac magnetic resonance imaging are diagnostic tools that detect cardiovascular complications of patients with inflammatory rheumatic diseases, often leading to prognosis. At the same time, their results can provide additive risk stratification for asymptomatic patients within the primary prevention setting [64]. Several studies have shown the value of novel echocardiography methods (Tissue Doppler Imaging), and most importantly global longitudinal strain by speckle tracking to asses left ventricular diastolic dysfunction and subclinical cardiac involvement [65,66,67]. A study conducted by Ikonomidis et al. that investigated the effects of interleukin-1 receptor antagonists on coronary and left ventricular function in rheumatoid arthritis patients with or without coronary artery disease, demonstrated that using these novel echocardiography methods is needed to reduce CV risk [68].

A table that summarized the results of this study and the percentage of different heart manifestations is listed below (Table 1).

## 5. Conclusions

Cardiac involvement due to systemic inflammation can occur in RA, SLE, SSc and AS with different prevalence and is frequently silent. Inflammatory rheumatic diseases can affect the myocardium, cardiac valves, pericardium, conduction system and arterial vasculature. Given the risk of increased cardiac mortality, disease prevention and surveillance are particularly important in these patients.

Early detection and adequate management of specific types of cardiac involvement are essential in inflammatory rheumatic diseases, and electrocardiographic and echocardiographic evaluations should be performed as routine investigations.

## Figures and Tables

**Figure 1 medicina-55-00249-f001:**
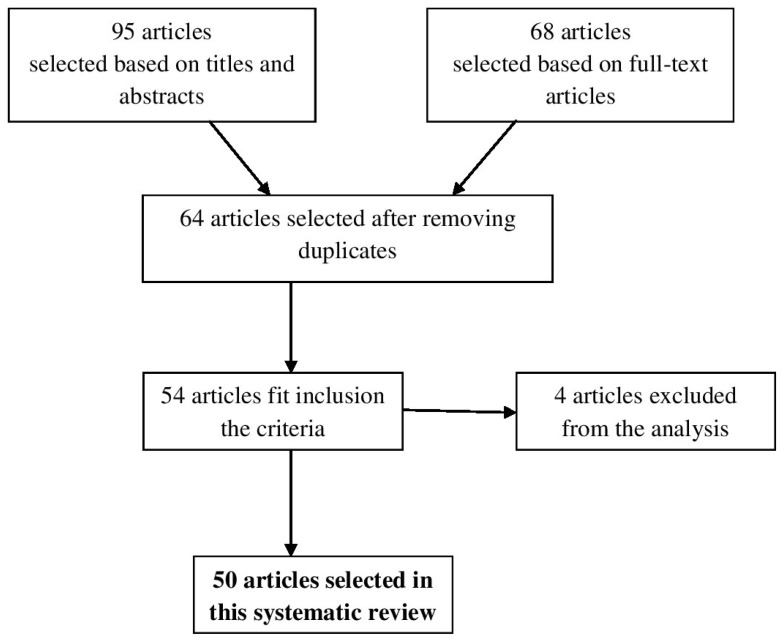
Diagram of search process.

**Table 1 medicina-55-00249-t001:** The heart manifestations in rheumatoid arthritis, systemic lupus erythematosus, systemic sclerosis and ankylosing spondylitis based on the findings of this study.

Cardiovascular Manifestations	Rheumatic Diseases	Findings
Electrical abnormalities	RA	Electric disorders were higher in RA patients compared to controls [6].
		Significant negative correlation was observed between disease activity and heart rate variability in RA patients [6].
		Right bundle branch block was found in 35% of 60 patients with RA [10].
		Repolarization defects and negative T waves occurred in 21% RA patients [12,13].
	SLE	Sinus tachycardia was reported in 50% of patients [6]. Malignant ventricular arrhythmias were rarely reported in SLE patients.
		Conduction defects appeared as a sequel of myocarditis in 34–70% SLE patients [6].
		CHB in SLE adults with anti-Ro/La antibodies was reported in 11 cases [16].
	SSc	Atrial fibrillation, flutter or paroxysmal supraventricular tachycardia were reported in 20–30% of patients. Up to 67% SSc patients had ventricular arrhythmias [6].
		25–75% of patients registered bundle and fascicular blocks, and very rare second- and third-degree AV block (<2%) [6].
		The most common arrhythmia was premature ventricular contraction, which is associated with a risk of 50% mortality and SCD [6].
	AS	2–20% of AS patients registered conduction disturbances. First-degree AV block were most common. Higher grade block, right and left bundle branch block were reported [17].
Valvular involvement	RA	30% of patients with RA presented valvular diseases. Mitral regurgitation was found in 80% of patients with RA [17].
	SLE	More than 50% of patients had valvular abnormalities in SLE [3]. Libman–Sacks endocarditis was more commonly detected in SLE patients [20].
	SSc	Aortic and mitral valves with regurgitation were found in 18% of autopsied SSc patients [22,23].
	AS	Valvular abnormalities described in AS were: aortic root thickening and dilatation; aortic cusp thickening and retraction; and aortic and mitral regurgitation [18]. Aortic root thickening was more common than dilatation (61% vs. 25%) [25].
Myocardial involvement	RA	Myocardial disease was rare among RA patients. The risk of myocardial dysfunction and CHF were high in RA patients compared to controls [26,27].
	SSc	Cardiac manifestations were reported in 15–35% of SSc patients. Myocardial fibrosis was the most common cardiac manifestation of SSc [31].
		Pulmonary involvement (interstitial fibrosis and pulmonary vascular disease) was detected in patients with SSc. This led to pulmonary arterial hypertension and associated myocardial changes [32].
	AS	The risk of heart failure was found to be 1.34-fold greater, considering the cause of LV diastolic dysfunction [35].
Pericardial involvement	RA	Between 30% and 50% of patients with RA had pericarditis. Clinically <10% of patients were diagnosed with severe RA [5,37].
	SLE	Pericardial disease occurred in 20–50% of SLE patients [18,38]. Moderate to large pericardial effusions were detected in 7% of patients in one series of SLE patients [32].
	SSc	Pericardial involvement (fibrinous pericarditis, chronic fibrous pericarditis, pericardial adhesions and pericardial effusions) occurred in 33–72% of SSc patients [23].
Vascular involvement	RA	Premature and accelerated atherosclerosis was detected in RA patients [42,44].Increased frequency of myocardial infarction in patients with RA was described when compared to controls [45].
	SLE	Premature atherosclerosis occurred in lupus patients [3].
	SSc	SSc was marked by microvascular abnormalities, secondary ischemia and excessive fibroblast activity. Involvement of large arteries was also reported [53].
	AS	CAD risk in patients with AS vs. controls was 1:41, demonstrating an increased risk [54].

RA: rheumatoid arthritis, SLE: systemic lupus erythematosus, SSc: systemic sclerosis, AS: ankylosing spondylitis, CHB: congenital heart block, AV: atrioventricular, SCD: sudden cardiac death, CHF: congestive heart failure, CAD: coronary artery disease.

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
