# Peer review of "Heart Involvement in Inflammatory Rheumatic Diseases: A Systematic Literature Review"

_medicina, 2019, doi:10.3390/medicina55060249_

Round 1
Reviewer 1 Report
Cardiovascular disease is increased in patients with inflammatory arthritis and connective-tissue diseases. The authors reviewed this issue and provided data supporting the increased risk of cardiovascular disease in patients with rheumatic disease.
I have suggested a number of comments that from my point of view may enhance the quality of this review.
Specific points:
In the Introduction you have to include a recent review article that focused on the epidemiology, traditional cardiovascular risk factors, genetics, and the interaction between chronic inflammation, atherosclerosis, and cardiovascular disease in inflammatory arthritis. In this this review article the authors also discussed controversial issues on the assessment of the cardiovascular risk in patients with inflammatory arthritis to better identify high-cardiovascular risk patients with inflammatory arthritis (Ref. Castañeda S, Nurmohamed MT, González-Gay MA. Cardiovascular disease in inflammatory rheumatic diseases. Best Pract Res Clin Rheumatol. 2016
Oct;30(5):851-869).
Introduction: Regarding the genetic component: Replace Reference 2 with a recent one that addressed in depth this issue: López-MejÃas R, Castañeda S, González-Juanatey C, Corrales A, Ferraz-Amaro I, Genre F, Remuzgo-MartÃnez S, Rodriguez-Rodriguez L, Blanco R, Llorca J, MartÃn J, González-Gay MA. Cardiovascular risk assessment in patients with rheumatoid arthritis: The relevance of clinical, genetic and serological markers. Autoimmun Rev. 2016 Nov;15(11):1013-1030. doi: 10.1016.
Material and Methods: In this section the authors say that the search was narrowed to 2005-2015. In contrast, in the summary (abstract) it is said that the assessment mainly encompassed the period 2007-2017. Please, clarify this point.
Discussion: Patients with rheumatoid arthritis have left ventricular diastolic dysfunction that may a predisposing factor for heart failure. This was found to be independent of the presence of traditional cardiovascular risk factors and it is probably related to a chronic proinflammatory state. Extra-articular manifestations were more common in patients with rheumatoid arthritis and diastolic dysfunction. Please, include in your review this information, adding the corresponding reference (Gonzalez-Juanatey C, Testa A, Garcia-Castelo A, Garcia-Porrua C, Llorca J, Ollier WE, Gonzalez-Gay MA. Echocardiographic and Doppler findings in long-term treated rheumatoid arthritis patients without clinically evident cardiovascular disease. Semin Arthritis Rheum. 2004 Feb;33(4):231-8. PubMed PMID: 14978661).
Vascular involvement in rheumatoid arthritis: The authors are entirely right when they say that premature atherosclerosis in patients with rheumatoid arthritis may occur in absence of traditional cardiovascular risk factors. Please, include the first study showing increased prevalence of severe subclinical atherosclerotic findings in long-term treated rheumatoid arthritis patients without clinically evident atherosclerotic disease (Ref. Gonzalez-Juanatey C, Llorca J, Testa A, Revuelta J, Garcia-Porrua C, Gonzalez-Gay MA. Increased prevalence of severe subclinical atherosclerotic findings in long-term treated rheumatoid arthritis patients without clinically evident atherosclerotic disease. Medicine (Baltimore). 2003 Nov;82(6):407-13. PubMed PMID: 14663290).
Discussion: Lines 276-279: The authors say that endothelial dysfunction, carotid intima-media thickness and plaque evaluations provide accurate detection of atherosclerotic process at a preclinical stage, before appearance of clinical disease allowing preventive measure introduction in order to decrease the cardiovascular risk in subjects with RA [44]. The authors should also include a review article that dealt in depth with this issue (Add the following Reference: Gonzalez-Gay MA, Gonzalez-Juanatey C, Vazquez-Rodriguez TR, Martin J, Llorca J. Endothelial dysfunction, carotid intima-media thickness, and accelerated atherosclerosis in rheumatoid arthritis. Semin Arthritis Rheum. 2008 Oct;38(2):67-70. doi: 10.1016).
In line with the above, I suggest the authors emphasize that risk charts algorithms used for the cardiovascular risk stratification, such as the SCORE or Framingham, underestimate the actual cardiovascular risk of patients with inflammatory arthritis. In this regard, non-invasive surrogate markers such as Coronary Artery Calcification Score (CACS) and carotid ultrasonography identified high cardiovascular risk rheumatoid arthritis or ankylosing spondylitis patients that had been included in the category of moderate cardiovascular risk when the SCORE or the modified EULAR SCORE were applied. Add the following references (Ref. A, Ref. B and Ref. C) to support your statement:
Ref. A. Corrales A, Parra JA, González-Juanatey C, Rueda-Gotor J, Blanco R, Llorca J,
González-Gay MA. Cardiovascular risk stratification in rheumatic diseases: carotid ultrasound is more sensitive than Coronary Artery Calcification Score to detect subclinical atherosclerosis in patients with rheumatoid arthritis. Ann Rheum Dis. 2013 Nov;72(11):1764-70. doi: 10.1136/annrheumdis-2013-203688. Epub 2013 Jul 13. PubMed PMID: 23852762.
Ref. B. Corrales A, González-Juanatey C, Peiró ME, Blanco R, Llorca J, González-Gay MA. Carotid ultrasound is useful for the cardiovascular risk stratification of patients with rheumatoid arthritis: results of a population-based study. Ann Rheum Dis. 2014 Apr;73(4):722-7. doi: 10.1136/annrheumdis-2012-203101. Epub 2013 Mar 16. PubMed PMID: 23505241.
Ref. C. Rueda-Gotor J, Llorca J, Corrales A, Parra JA, Portilla V, Genre F, Blanco R,
Agudo M, Fuentevilla P, Expósito R, Mata C, Pina T, González-Juanatey C, González-Gay MA. Cardiovascular risk stratification in axial spondyloarthritis: carotid ultrasound is more sensitive than coronary artery calcification score to detect high-cardiovascular risk axial spondyloarthritis patients. Clin Exp Rheumatol. 2018 Jan-Feb;36(1):73-80. Epub 2017 Aug 28. PubMed PMID: 28850022.
Author Response
Response to Reviewer 1:
1) In the Introduction we included a suggested review article that focused on the epidemiology, traditional cardiovascular risk factors, genetics, and the interaction between chronic inflammation, atherosclerosis, and cardiovascular disease in inflammatory arthritis (Ref. 4: Castañeda S, Nurmohamed MT, González-Gay MA. Cardiovascular disease in inflammatory rheumatic diseases. Best Pract Res Clin Rheumatol. 2016;30(5):851-869).
2) We replaced Reeference 2 with a recent one: López-MejÃas R, Castañeda S, González-Juanatey C, Corrales A, Ferraz-Amaro I, Genre F, Remuzgo-MartÃnez S, Rodriguez-Rodriguez L, Blanco R, Llorca J, MartÃn J, González-Gay MA. Cardiovascular risk assessment in patients with rheumatoid arthritis: The relevance of clinical, genetic and serological markers. Autoimmun Rev. 2016;15(11):1013-1030.
3) Material and Methods: The search was narrowed to 2002-2018. The same period has been place in abstract section.
4) Discussion: Left ventricular diastolic dysfunction in rheumatoid arthritis patients was found to be independent of the presence of traditional cardiovascular risk factors and it is probably related to a chronic proinflammatory state. Extra-articular manifestations were more common in patients with rheumatoid arthritis and diastolic dysfunction. Thus we added Ref. 27 (Gonzalez-Juanatey C, Testa A, Garcia-Castelo A, Garcia-Porrua C, Llorca J, Ollier WE, Gonzalez-Gay MA. Echocardiographic and Doppler findings in long-term treated rheumatoid arthritis patients without clinically evident cardiovascular disease. Semin Arthritis Rheum. 2004;33(4):231-8).
5) Discussion: An increased prevalence of severe subclinical atherosclerotic findings in long-term treated rheumatoid arthritis patients without clinically evident atherosclerotic disease was reported for the first time by Gonzalez-Juanatey C et al. [48].
6) Discussion: Endothelial dysfunction, carotid intima-media thickness and plaque evaluations provide accurate detection of atherosclerotic process at a preclinical stage, before appearance of clinical disease allowing preventive measure introduction in order to decrease the cardiovascular risk in subjects with RA [ 46, 47]. We added reference 47.
7) Discussion
According to the current ESC (European society of cardiology) guidelines, cardiovascular risk quantification in these patients is recommended every 5 years for patients with low to moderate cardiovascular risk, because there is currently no evidence that its assessment each year for these patients reduces cardiovascular risk more than screening every 5 years. In intermediary and high cardiovascular risk patients, screening should be performed more often in order to rapidly modify and lowering existing risk factors. The new EULAR Guideline Recommendation includes the option of screening for carotid atherosclerotic plaques, especially in RA patients due to the increased association with CAD in these patients. [Ref 55,56].
In studies conducted on patients with inflammatory rheumatic diseases it was observed that the risk charts algorithms used for the cardiovascular risk stratification, such as the risk SCORE (Systematic Coronary Risk Evaluation) or Framingham, underestimate the actual cardiovascular risk of patients with these diseases. In this regard, non-invasive surrogate markers such as Coronary Artery Calcification Score (CACS) and carotid ultrasonography identified high cardiovascular risk rheumatoid arthritis or ankylosing spondylitis patients that had been included in the category of moderate cardiovascular risk when the SCORE or the modified EULAR SCORE (the European League Against Rheumatism ) were applied [Ref 57-59].
We added the suggested references.
Reviewer 2 Report
Dear authors,
in this paper the author addressed the role of heart in rheumatic disease. The paper is interesting, but I think it is lacking in some parts, and a critical revision for clinical suggestions could be helpful.
First of all there is a mistakes on years of internet research (2005-2015 or.... 2017?).
There is no mention in
Electrical abnormalities are represented by sudden cardiac death, ventricular arrhythmia,
89 supraventricular tachycardia and atrioventricular block, predominantly affecting the patients with
90 systemic sclerosis, rheumatoid arthritis, and ankylosing spondilitis (5)
and SLE that after you discussed?
A table that summarized the findings of this review and the percentage of involvement of different heart manifestations could be useful.
For rheuamtoid arthritis and systemic sclerosis there are report on very early heart involvement in the disease, please discuss.
A final table or paragraph with clinical suggestions considering the different risk of these 4 rheumatic diseases to categorize the risk of patients and to decide how frequent the heart follow-up will be performed.
Author Response
Response to Reviewer 2:
1) The search was conducted from 2002-2018. The same period has been placed in the abstract section.
2) Electrical abnormalities are represented by sudden cardiac death, ventricular arrhythmia, supraventricular tachycardia and atrioventricular block, predominantly affecting the patients with systemic sclerosis, rheumatoid arthritis, systemic lupus erythematosus and ankylosing spondilitis [6].
3) A table that summarized the findings of this review with the percentage of cardiac manifestations was added (Table I).
4) We added a final paragraph with the cardiovascular risk of rheumatic patients and how frequent the heart follow-up should be performed.
According to the current ESC (European society of cardiology) guidelines, cardiovascular risk quantification in these patients is recommended every 5 years for patients with low to moderate cardiovascular risk, because there is currently no evidence that its assessment each year for these patients reduces cardiovascular risk more than screening every 5 years. In intermediary and high cardiovascular risk patients, screening should be performed more often in order to rapidly modify and lowering existing risk factors. The new EULAR Guideline Recommendation includes the option of screening for carotid atherosclerotic plaques, especially in RA patients due to the increased association with CAD in these patients. [Ref 55,56].
In studies conducted on patients with inflammatory rheumatic diseases it was observed that the risk charts algorithms used for the cardiovascular risk stratification, such as the risk SCORE (Systematic Coronary Risk Evaluation) or Framingham, underestimate the actual cardiovascular risk of patients with these diseases. In this regard, non-invasive surrogate markers such as Coronary Artery Calcification Score (CACS) and carotid ultrasonography identified high cardiovascular risk rheumatoid arthritis or ankylosing spondylitis patients that had been included in the category of moderate cardiovascular risk when the SCORE or the modified EULAR SCORE (the European League Against Rheumatism ) were applied [Ref 57-59].
Round 2
Reviewer 1 Report
The quality of the manuscript has been considerably improved.
I congratulate the authors on this elegant piece of work..
Author Response
All revisions are clearly highlighted with the "Track Changes" function in the Microsoft Word document, so that they are easily visible. We add references 62-68.
Below you will find our response:
The last European Guidelines on cardiovascular disease prevention in clinical practice, recommend to multiply the risk SCORE with 1.5 to assess the CV risk in patients with autoimmune disease, because the actual risk is underestimated in these patients [62, 63]. Statin trials suggest that the relative reduction in CVD incidence in patients with autoimmune diseases sustain that these drugs are indicated when the risk SCORE ≥10% (very-high-risk persons) [62].
Carotid ultrasound, echocardiography, cardiac computed tomography, positron emission tomography, and cardiac magnetic resonance imaging are diagnostic tools that detect cardiovascular complications of patients with inflammatory rheumatic diseases, often leading to prognosis. At the same time, their results can provide additive risk stratification for asymptomatic patients within the primary prevention setting [64]. Several studies have shown the value of novel echocardiography methods (Tissue Doppler Imaging ), and most importantly global longitudinal strain by speckle tracking to asses left ventricular diastolic dysfunction and subclinical cardiac involvement [ 65, 66,67]. In a study conducted by Ikonomidis et al, that investigated the effects of interleukin-1 receptor antagonists on coronary and left ventricular function in rheumatoid arthritis patients with or without coronary artery disease using these novel echocardiography methods, demonstrated that this treatment is needed to reduce CV risk [68].